# The Role of GH/IGF Axis in Dento-Alveolar Complex from Development to Aging and Therapeutics: A Narrative Review

**DOI:** 10.3390/cells10051181

**Published:** 2021-05-12

**Authors:** Kouassi Armel Koffi, Sophie Doublier, Jean-Marc Ricort, Sylvie Babajko, Ali Nassif, Juliane Isaac

**Affiliations:** 1Centre de Recherche des Cordeliers, Lab. of Molecular Oral Pathophysiology, Université de Paris, Sorbonne Université, INSERM, 75006 Paris, France; armelk_ioscocody@yahoo.fr (K.A.K.); sophie.doublier@sorbonne-universite.fr (S.D.); jean-marc.ricort@ens-paris-saclay.fr (J.-M.R.); dent4world@gmail.com (A.N.); juliane.isaac@u-paris.fr (J.I.); 2Department of Oral Biology, Dental Faculty, Université de Paris, 75006 Paris, France; 3Ecole Normale Supérieure Paris-Saclay, Université Paris-Saclay, 91290 Gif-sur-Yvette, France; 4Service d’Orthopédie Dento-Faciale, Sites Hospitaliers Pitié Salpêtrière et Rothschild, Centre de Référence Maladies Rares Orales et Dentaires (O-Rares), AP-HP, 75012 Paris, France

**Keywords:** GH, IGF, IGFBP, animal models, age-related pathologies, tooth, enamel, dentin, dental ligament, alveolar bone, mandible, oral tissue engineering, dentin repair, orthodontic treatment, periodontal regeneration, osseointegration, bone repair

## Abstract

The GH/IGF axis is a major regulator of bone formation and resorption and is essential to the achievement of normal skeleton growth and homeostasis. Beyond its key role in bone physiology, the GH/IGF axis has also major pleiotropic endocrine and autocrine/paracrine effects on mineralized tissues throughout life. This article aims to review the literature on GH, IGFs, IGF binding proteins, and their respective receptors in dental tissues, both epithelium (enamel) and mesenchyme (dentin, pulp, and tooth-supporting periodontium). The present review re-examines and refines the expression of the elements of the GH/IGF axis in oral tissues and their in vivo and in vitro mechanisms of action in different mineralizing cell types of the dento-alveolar complex including ameloblasts, odontoblasts, pulp cells, cementoblasts, periodontal ligament cells, and jaw osteoblasts focusing on cell-specific activities. Together, these data emphasize the determinant role of the GH/IGF axis in physiological and pathological development, morphometry, and aging of the teeth, the periodontium, and oral bones in humans, rodents, and other vertebrates. These advancements in oral biology have elicited an enormous interest among investigators to translate the fundamental discoveries on the GH/IGF axis into innovative strategies for targeted oral tissue therapies with local treatments, associated or not with materials, for orthodontics and the repair and regeneration of the dento-alveolar complex and oral bones.

## 1. Introduction

GH/IGF axis or somatotropic axis consists of growth hormone (GH), insulin-like growth factors (IGFs), their receptors (GHR and IGF-1R, respectively), and IGF binding proteins (IGFBPs). GH is secreted by the anterior pituitary gland and acts directly on target tissues by its specific receptor or indirectly by up-regulating the production of IGF1 in the liver and in other target tissues. IGFs transmit their effects by IGF-1R, which is expressed in most tissues including bone and dental cells. The somatotropic axis is known for its crucial role in postnatal growth and development. More specifically, the GH/IGF axis has a strong influence on the growth and metabolism of craniofacial bones and dental tissues reviewed here.

Clinical observations report that many patients with GH dysregulations who are suffering from dwarfism or acromegaly, also have tooth and cranio-facial bone dysmorphology underlying a particular role of the GH/IGF axis in facial mineralized tissues (see [1] for review). Analysis of GHR variant phenotypes showed an association between P561T variant and the mandibular length as well as a lower face height supporting the GHR that might be a candidate gene for mandibular morphogenesis [2]. While GH/IGF/IGFBP endocrine and paracrine actions in bone are well documented for growth and ageing in axial and appendicular skeleton, the effects of these molecules have been much less investigated in the cranio-facial bones, and even less in the dento-alveolar complex where dental and bone cells are interacting (see [1,3,4,5,6] for reviews). Indeed, bone cells present specificities depending on their localization site and embryonic origin, as already reported by Kasperk et al., who showed higher mRNA levels for IGF2 in human mandibular osteoblastic cells compared to iliac osteoblastic cells [7].

Expression of the GHR by most mineralizing cells in cranio-facial bones and dental cells argue for a direct physiological role of GH in these tissues without excluding some IGF effects transmitted by the widely expressed IGF-1R and modulated by IGFBPs, that may also have additional IGF-independent effects. Among the various hormones and growth factors investigated, GH appears to be one of the most closely associated with dental maturity [8]. Recent studies have shown that GHR variants (rs6184 and rs6180) are associated with specific tooth and root dimensions [9,10], as well as with mandibular morphology [11]. Genetic polymorphisms in GHR (rs1509460) are also associated with developmental defects of enamel arguing for the GHR contribution to dental enamel synthesis and structure [12].

These data demonstrate the involvement of GH and GHR in physiological development of the dento-alveolar complex in addition to their well-known consequences when GH levels are deregulated notably on mandibular growth (prognathism or retrognathism). However, the beneficial or detrimental consequences of deregulated GH secretion on dento-alveolar complex are controversial, as reported recently for acromegaly, which does not appear to induce generalized hypertrophy of the gingiva or hypercementosis, but could protect from a severe periodontal disease by conferring more robust periodontal tissues [10,13] due to an increase in the levels of protective molecules such as bone morphogenetic protein 2 (BMP2) [14].

In addition to the presentation of current knowledge on GH and IGF/IGFBP expressions and their actions in oral and facial mineralized tissues (excluding chondrocytes and cartilage), this review also focuses on the data reporting the use of recombinant human GH (rhGH) and rhIGF1 as therapeutic tools for targeted oral tissue therapies with local treatments, associated or not with materials, for orthodontic movements or the repair and regeneration of the dentin and oral bones.

## 2. Search Strategy

A search of the literature was conducted until 30 April 2021. The eligibility criteria were as follows: (a) original in vitro, in vivo, or preclinical and clinical studies; (b) systematic and narrative reviews; (c) abstracts available in English or French. Letters to editors, case series, commentaries, conferences abstracts, and case reports were excluded. Online database of PubMed was searched using a combination of the terms insulin like growth factor (IGF), growth hormone (GH), enamel, dentin, dental ligament, alveolar bone, mandible, skull, oral bone, ameloblast, odontoblast, cementoblast, alveolar osteoblast, and oral osteoblast. Studies that were not directly related to oral biology and mineralized tissues, or studies on cartilages (Meckel cartilage, condyle) and/or chondrocytes were excluded. An initial assessment based on the reading of titles and abstracts was performed by at least two authors for each study. Any discrepancies in the inclusion of an article were resolved by discussion among the authors. A total of 2.104 articles were identified, of which 534 articles including 18 reviews fulfilled our eligibility criteria and were considered by the authors as relevant for the topic. A screening of full texts of these articles was next performed by the authors to include in the present review the pivotal original studies and reviews focusing on the specific expression and actions of GH and IGFs/IGFBPs in oral and facial mineralized tissues. Finally, 160 articles were included in the review, 141 selected from the PubMed search and 19 describing the dento-alveolar complex and its tissue therapies.

## 3. Expression and Action of GH/IGF Axis in the Dento-Alveolar Complex

First, a brief introduction of all the tissues of the dento-alveolar complex will help to better understand the interactions between the different types of cells involving GH and IGFs/IGFBPs. The dento-alveolar complex is constituted by the teeth attached to the underlying alveolar bone through the cement and the periodontal ligament (PDL) (Figure 1). The teeth are formed after initial epithelial cell proliferation and interaction with the oral mesenchyme in the place of dental lamina that determines the number of teeth. Dental development is characterized by specific stages called placodes, bud, cap, and bell (Figure 1A) that finally form the teeth with the crown corresponding to the erupted part, and the root anchored in the alveolar bone. The most external layer of the crown is the enamel that is the most mineralized tissue in living organisms, consisting of 96–97% apatite crystals and only 3% of organic matter and water. It is an avascular and non-innervated matrix. Enamel is synthesized by ameloblasts, which are epithelial cells able to change their shape and functions during the process of amelogenesis (Figure 1B) (see [15] for review). Despite important differences in dentition in mammals, the amelogenesis process is very similar with the specificity of rodent continually growing incisor that concomitantly exhibits all the stages and thus constitutes a model of choice for studies of amelogenesis (Figure 1B). First, precursor cells are proliferating in the cervical loop, then are committed and differentiate into secreting ameloblasts that secrete Enamel Matrix Proteins (EMPs), mainly amelogenins and enamelin, crucial for enamel mineralization initiation (Figure 1B and 1C). Contrary to bone mineralization, collagen 1α1 is not involved in enamel mineralization, and EMPs are degraded by proteases, mainly MMP20 and KLK4, for complete enamel mineralization (see [15,16] for reviews). The degradation of EMPs and the enamel terminal mineralization are achieved by maturation-stage ameloblasts that secrete high levels of proteases, present high levels of ion pumps and transporters at their apical side with a fine-tuned pH regulation that allow the assembly of apatite crystals for complete enamel mineralization [15]. The two main stages of amelogenesis, secretion- and maturation-stages, are characterized by specific ameloblasts with particular cell features that allow them to change their shape and function during the short and brutal transition stage. At the end of amelogenesis, ameloblasts are lost, leading to an acellular and irreparable enamel matrix. Epithelial cells localized in the radical part of the teeth form the Hertwig’s epithelial root sheath (HERS)/Malassez rest and contribute to the formation of the periodontal ligament with the cementoblasts that participate to anchor the tooth to the alveolar bone (Figure 1). Alveolar bone presents many specificities that distinguish it from all other bones (see [17] for review). Its formation is tightly associated with dental development and eruption, which means that its size, structure, and shape are dependent on teeth [18] and it may disappear when the teeth are lost. The alveolar bone protects the dental root whose development begins after the crown formation. The dental root elongation tightly involves the osteoblasts and osteoclasts of the alveolar bone, the odontoblasts and the dental pulp cells. Odontoblasts are cells in charge of the dentin formation, which underlies the enamel in the coronal part and the cement in the radicular part. The dentin protects the pulp, which ensures the vitality of the dental organ. Odontoblasts are mesenchymal cells whose differentiation is coordinated with that of ameloblasts during tooth formation to ensure coordinated dentin and enamel formation (Figure 1) (see [19] for review). Contrary to enamel forming cells that are lost after tooth eruption, the odontoblasts are present throughout life and may synthesize tertiary dentin matrix (reactionary or reparative) beneath the site of injury. Of note, the dental pulp contains the dental pulp stem cells (DPSCs), as well as the apical zone of the formed dental root contains the stem cells from apical papilla (SCAPs) (see [20] for review). Together, DPSCs and SCAPs constitute an important reservoir of stem cells in living teeth with promising properties in tissue repair and regeneration due to their ability to differentiate into epithelial and mesenchymal cells [21,22,23,24]).

Interestingly, studies report that IGF2 secreted by mesenchymal cells induces the expression of IGF-1R in mouse epithelial cell 3D cultures [25], thus contributing to the coordinated differentiation process of odontoblasts and ameloblasts. The microdissected medial part of the mouse mandibular arch of E10.5 embryos showed that IGF1 was highly expressed in the mesenchyme, IGF2 and IGF-1R were expressed in both the midline epithelium and surrounding mesenchyme, and IGFBP5 was highly expressed in the epithelium [26].

The knowledge of factors driving the development of the dento-alveolar complex and being involved in cell differentiation is crucial for the development of innovative therapeutic strategies. Among all the factors involved, the GH/IGF axis seems to play an important role in all tissues of the dento-alveolar complex.

### 3.1. Dental Epithelium and Enamel

GHR and IGF1 proteins are detected in the incisor and first molar of rat embryos with high signals in dental epithelial cells, suggesting the involvement of GH/IGF axis in dental development [27,28]. In addition, GH treatments enhance IGF1 expression in dental epithelial cells demonstrating their responsiveness to this hormone [29]. IGF1 and IGF-1R involvement in amelogenesis was confirmed when Joseph et al. showed a relationship between a decrease in IGF-1R expression and ameloblasts apoptosis during two key steps of amelogenesis, the transition period when ameloblasts suddenly change their shape and their function, between the secretion- and the maturation-stages, and at the end of the maturation-stage just before tooth eruption [30] (Figure 2A,B).

In the continuously erupting rat incisor, IGF1, IGF2, IGF-1R proteins and mRNAs are detected in ameloblasts with strong signal in undifferentiated cells of the cervical loop, lower signals in secreting ameloblasts, and higher again in the maturation-stage ameloblasts in charge of enamel terminal mineralization [31]. More precisely, IGF1 and IGF2 mRNAs are detected preferentially in maturation-stage ameloblasts, with a preferential expression in ruffle-ended ameloblasts that are involved in calcium transport and pH regulation, two important parameters for enamel mineralization (see [15] for review). Similarly, in mouse, the first mandibular molars were dissected from E16 and E17 mouse embryos and placed in organ culture. Most mRNAs coding IGF system elements except IGFBP6 and probably IGFBP1 were detected by RT-PCR [32]. In this model, IGF1 treatment induces enamel matrix proteins, amelogenins, and enamelin responsible for the depth of enamel.

IGFs and IGF-1R expression during the development of human incisor tooth germs between the 7th and the 20th week of development was also investigated [33]. IGF2 is highly expressed mostly in the cervical loop of the enamel organ in highly proliferative cells and in differentiating pre-ameloblasts and pre-odontoblasts of the cusp tip region during the early and late bell stages when enamel organ acquires definitive shape. Expression patterns of investigated IGF system elements are time- and space-dependent indicators of the importance of these factors in crown morphogenesis of human incisors.

### 3.2. Dental Mesenchyme and Dentin

In various mouse transgenic models, GH status was found to influence the crown width, the root length, and the dentin thickness [34]. This is concordant with the expression of paracrine GH and GH receptors during tooth bud morphogenesis, and of GH receptors in the enamel organ, dental papilla, and Hertwig’s epithelial root sheath (HERS) during dentinogenesis. Based on prior studies carried out in mineralizing cells, these GH morphogenetic actions may be mediated by the induction of both bone morphogenetic proteins (essentially BMP2 and BMP4) and IGFs.

IGFBP2 is expressed in mouse dental mesenchyme but not in epithelial cells. IGFBP2, whose expression is down-regulated by the Runt-related transcription factor 2 (RUNX2) master gene, is proposed to actively contribute to the pathophysiology of odontoblasts and pulp cells [35]. RUNX2 expression is upregulated by IGF1 in osteoblasts and odontoblasts [36]. In addition to IGFBP2 and IGF1, predentin and odontoblastic processes are also stained for IGFBP3 [37]. The detailed expression patterns of IGF1, IGF-1R, IGFBP3, and IGFBP5 were examined in the mouse incisor mesenchyme [38]. IGF1 and IGF-1R are found mainly in undifferentiated dental papilla cells and preodontoblasts whereas IGFBP3 and IGFBP5 are expressed in more differentiated odontoblasts (Figure 2B). The authors concluded that IGFBP3 regulates the transition from the proliferative to differentiation stage by inhibiting the action of IGF1 on the proliferation of dental papilla cells, and that IGFBP5 plays an important role in the maintenance of the differentiated odontoblasts during tooth development.

### 3.3. Dental Pulp

The dental pulp contains several types of cells (e.g., fibroblasts, immune cells, DPSCs) embedded in the extracellular matrix (ECM) containing IGF system components. Human and rodent dental pulp cells (DPCs) have been shown to express all elements of the IGF system (Figure 2B).

In freshly extracted human third molars, IGF-1R presents a greater expression in pulp cells of teeth having incomplete root development [39], whereas IGF1 is preferentially expressed in dental pulp of teeth with complete root development [40]. Authors hypothesized that pulp cell proliferation is constant and not dependent on the root development stage to maintain the tolerance and the capacity of the pulp to respond to physical, chemical or mechanical aggression. This proliferative ability allows pulp cells to constantly renew themselves, maintain tissue homeostasis, or form new hard tissue as a defense mechanism.

IGFBPs are secreted by DPCs as shown by Götz et al., who detected all IGF components including IGFBP1 in human teeth [37]. This was not completely confirmed by Al-Khafaji et al., who did not detect IGFBP1, but found IGFBP4 as one of the main IGFBP expressed by DPCs [41]. Others found that DPCs express IGF1, IGFBP1, IGFBP3, IGFBP5, and IGFBP6 [42].

IGFBP3, whose expression and secretion are coordinated with that of IGFBP2, regulates the transition from the proliferative to the differentiation stage by inhibiting the action of IGF1 on the proliferation of human DPCs [43]. IGFBP5 plays an important role in the maintenance of the differentiated odontoblasts [38]. IGF-independent action of IGFBP5 might play a key role in the regulation of cell survival and apoptosis in dental pulp stem/progenitor cells following tooth injury [44]. However, the precise variations of each IGFBP levels and their corresponding roles in pathophysiological processes remain discussed as IGFBP may have IGF-dependent and -independent functions. Moreover, their limited proteolysis may either enhance or inhibit IGF functions leading to apparent controversial conclusions depending on the studies. Thus, further investigations are required to understand their roles and their possible use in innovative therapeutic strategies.

### 3.4. Cement and Periodontal Ligament

The periodontal ligament is constituted by different types of cells, cementoblasts sharing many features with odontoblasts and osteoblasts, epithelial rests of Malassez forming the Hertwig’s epithelial root sheath (HERS), fibroblasts, and periodontal ligament stem cells (PDLSCs) (Figure 1). HERS and the apical papilla are two embryonic structures that coordinate the entire radicular development through multiple epithelial-mesenchymal interactions. Although cementum has been poorly studied until now, the similarities between cementoblasts/cementocytes and osteoblasts/osteocytes and the numerous stresses it undergoes (mechanical forces, orthodontic forces or periapical periodontitis) lead to hypothesizing an active remodeling activity in cementum that may share similar cellular mechanisms with the Bone Remodeling Compartment (BRC). In that context, Brochado Martins et al. recently provided histological evidence of a specialized remodeling compartment in root cementum [45].

In rats, cementoblasts and odontoblasts localized at sites of new matrix formation show intense GHR immunoreactivity, whereas mature cementoblasts and odontoblasts at later stages of tooth development are nonreactive [46]. These patterns of GHR expression during odontogenesis suggest that GH may promote the functional state of these cells. The importance of GH on cellular cementum length was confirmed by Smid et al., who proposed therapeutic applications of GH to help regeneration of the periodontium based on its high responsiveness to GH [47]. The cementoblasts, osteoblasts, and PDL cells (PDLCs) responded to GH by expressing osteogenic markers, BMP2 and BMP4, BMPR1A, alkaline phosphatase (ALP), osteocalcin (OCN), and osteopontin (OPN), and by increasing the numbers of PDLCs [48]. However, while long-term treatment with GH may promote mineralization of human PDLCs and alveolar bone cells, short-term treatment does not promote proliferation of osteoblast precursors nor induce expression of late osteogenic markers [49]. IGF1, for its part, promotes not only proliferation, but also differentiation of human PDLSCs, by up-regulating the expression of RUNX2, SP7, and OCN, and by activating the phosphorylation of extracellular signal-regulated kinase (ERK) and c-Jun N-terminal kinase (JNK) arguing for large IGF1 effects transmitted by IGF-1R [50].

In human premolars, IGF-1R is expressed by periodontal fibroblasts. IGF1, IGF2, and the six IGFBPs could be detected by immunohistochemistry in the ECM of the adhering PDLCs, whereas only IGF2 could be detected in the acellular cementum [42,51]. Also of note, outer cementum layers with inserting Sharpey’s fibers reacted with all antibodies against the IGF system components except for IGFBP4 and IGFBP6. IGFBP6 and, to a lesser extent, IGFBP4 are expressed by epithelial rests of Malassez located in the periodontal ligament [51]. IGF1 was shown to induce elongation of HERS and increase cell proliferation in its outer layer illustrating IGF1 involvement in early root formation [52]. Considering the high affinity of IGFBP6 for IGF2, IGFBP6 may inhibit the mitogenic activity of IGFs present in the PDL on Malassez cells.

The PDL contains stem cells (PDLSCs) with interesting properties for tissue regeneration. Mouse periodontal ligament stem cells (PDLSCs) express IGFBP5 [53]. IGFBP5 administration activates PDLSCs and bone marrow mesenchymal stem cells (BMSCs) in vitro [54].

### 3.5. Alveolar Bone

In the alveolar bone of rats, osteoblasts engaged in intramembranous ossification and osteoclasts localized at sites of bone remodeling resorption are immunopositive for GHR, while osteocytes and endosteal cells are immunonegative [46]. Joseph et al. also reported a high IGF1 signal in osteoblasts and osteoclasts from rat alveolar bone [28]. These findings with those described above reporting high expression of IGFs in PDL and HERS support the notion of paracrine or autocrine functions of IGF1 in dental root development [52].

During bone formation, growth factors (including IGF1) released from the bone matrix during osteoclastic bone resorption stimulate osteoblast differentiation, thus bone remodeling. Molar root formation and tooth eruption are dependent on both (anabolic) osteoblast and (catabolic) osteoclast activities controlled by receptor activator of NF-κB ligand (RANKL) as demonstrated in RANKL -/- mice [55,56]. In these mutants, the IGF signaling pathway is down-regulated in cells involved in root elongation and this impairment is rescued by the addition of IGF1, demonstrating that dental root and eruption defect in RANKL mutant mice may result from failure of IGF1 release from bone matrix through osteoclast bone resorption.

## 4. Mechanisms of Action of the GH/IGFs/IGFBPs Axis on Dental and Bone Cells

Both GH and IGF1 exert their effects on osteogenic and more generally on mineralizing cells by binding to their receptor, leading to the activation of intracellular signaling pathways and the regulation of gene expressions that mediate cellular differentiation and function. To facilitate skeletal growth and metabolism, both GH and IGF1 work in concert with many other skeletal regulators, such as sex-steroids, thyroid hormones, and parathyroid hormone (see [57] for review). Many studies reported GH ability to stimulate osteoblast proliferation and differentiation as well as the expression of bone-specific genes such as collagen 1α1, OCN, ALP, BMP2, and BMP4. Crippa et al. investigated the effects of GH on osteoblasts isolated from patients at different ages and found donor-age-dependent effects of GH on the development of osteoblastic phenotype in cultures of cells from alveolar bone of adolescents (13–16 years old), young adults (18–35 years old), and adults (36–49 years old) [58]. Indeed, GH increases growth, collagen content and alkaline phosphatase (ALP) activity and upregulates mRNA levels of ALP, OCN, collagen 1α1, and RUNX2 in cultures from adolescents, whereas a non-significant effect is observed in cultures from adults indicating that GH effects are donor-age-dependent on both in vitro osteogenesis and gene expression of osteoblastic markers. Of note, GH not only stimulates osteogenic cells, but also osteoclastic bone resorption (see [59] for review). Locally produced IGF1 and IGFBP5 modulate the stimulation of bone remodeling by GH.

Although a direct and IGF-1R-independent action of GH on osteoblast apoptosis was demonstrated in vitro, IGF-1R is required for the anabolic effects of GH in osteoblasts in vivo [60]. The anabolic effects of IGFs on osteoblasts are modulated by several IGFBPs. All six IGFBPs are expressed with differential expression during osteoblast differentiation. IGFBP2 and IGFBP5 are maximally expressed by proliferating preosteoblasts, while mature osteoblasts showed a higher expression of IGFBP3, IGFBP4, and IGFBP6 (see [61] for review). The stimulatory effect of IGF1 may be increased by IGFBP3 and IGFBP5, whereas it is decreased by IGFBP4 [62]. In mice with a daily subcutaneous administration in calvaria of a rhIGF1/rhIGFBP5, the calvaria bone mass is clearly increased in a dose-dependent manner compared to treatments with IGF1 or IGFBP5 alone, suggesting a synergic effect of IGF1 and IGFBP5 on calvaria osteogenesis [63].

The GH/IGF axis at the tissue level is modulated by two possible mechanisms: liver and bone-derived IGF1 that inhibits pituitary GH secretion, and bone-derived IGF1 that inhibits local action of GH by reducing GHR availability [61]. Of note, IGFBPs also regulate both the number and the activity of GHR through modulation of IGF activity [61].

Despite the signaling pathways activated by GH, IGFs and IGFBPs were largely described in many different cell types, it is worth mentioning some recent works precisely described the signaling pathways activated by IGF1 in PDLSCs and alveolar osteoblasts [50,64]. Li et al. demonstrated that IGF1 with platelet-rich fibrin (PRF) promote the growth, proliferation, and differentiation of human PDLSCs by up-regulating the expression of RUNX2, SP7, and OCN [50]. Similarly, IGF1 in a naturally present mixture of growth factors (containing platelet-derived growth factor (PDGF-BB), transforming growth factor (TGFβ1), basic fibroblast growth factor (bFGF), and vascular endothelial growth factor (VEGF)) induces proliferation, migration, formation of mineralized nodules, and ALP, dentin sialophosphoprotein (DSPP), and dentin matrix acidic phosphoprotein (DMP1) expression in SCAPs, thus increasing their ability in osteogenic and odontogenic differentiation [65]. Interestingly, in vivo results revealed that IGF1-treated SCAPs synthetize bone-like tissues, while untreated SCAPs mainly generate dentin/pulp complex-like structures after transplantation [66]. Regarding the ligament cells, IGF1 substantially enhances survival of PDL fibroblasts compared with gingival fibroblasts by the up-regulation of anti-apoptotic molecules and the down-regulation of pro-apoptotic partners [67]. Furthermore, the differential expression of IGFBP5 in gingival fibroblasts and PDL cells observed in vivo and in vitro raises the question of IGFBP5 involvement in the PDL cell survival [67]. In oral cells as in most cells, IGF1 induces the phosphorylation of ERK and JNK and thus facilitates the proliferation and osteogenic differentiation of PDLSCs and SCAPs via the activation of the mitogen-activated protein kinase (MAPK) signaling pathway [50,68]. In human calvarial osteoblasts, there is a positive correlation between high IGF1 expression and expression of other osteogenic genes such as RUNX2 and a decrease in GSK3β, a serine/threonine kinase known to inhibit RUNX2, thus activating osteogenesis through the IRS1-mediated AKT pathway [64].

Recent data report that the IGF1/IGF-1R osteogenic/odontogenic activity is modulated at post-transcriptional level by specific long non-coding RNAs (lncRNAs) and microRNAs (miRs). Notably, Bian et al. investigated the role of the lncRNA (ANRIL) in the senescence and osteogenic differentiation of inflamed PDLSCs (iPDLSCs) isolated from periodontitis patients and healthy periodontal ligament stem cells (hPDLSCs) [69]. They observed that lncRNA ANRIL and IGF-1R were declined in iPDLSCs compared with hPDLSCs, while miR-7-5p was upregulated. Upregulating miR-7-5p inhibited the osteogenic differentiation of iPDLSCs and IGF-1R was identified as a direct target of miR-7-5p. Otherwise, Gan et al. showed that miR-221-3p and miR-222-3p were up-regulated in the mandibles of diabetic rats and BMSCs cultured in high glucose condition [70]. Silencing of miR-221-3p or/ and miR-222-3p increased ALP activity and up-regulated osteoblast-related protein levels. These effects are mediated by the IGF1/ERK signaling pathway as miR-221-3p and miR-222-3p both target IGF1 and cooperatively regulate its expression and IGF1-transmitted ERK activation. An additional study carried out in isolated human SCAPs showed that the miR let-7c, originally associated with osteogenic cells, is able to downregulate the expression of odonto/osteogenic markers (ALP, RUNX2, OSX, OCN, collagen 1α1, DSPP, and DMP1) [68]. Moreover, the JNK and p38 MAPK signaling pathways were activated in let-7c-low SCAPs, but inhibited in let-7c-over SCAPs.

Recent evidence, detailed in parts 6 and 7 of this review, confirmed the important role of GH and/or IGFs/IGFBPs in dental and bone cells and thus therapeutic treatments. For example, Bhattarai et al. reported the effects of IGFBP3 bound on chitosan gold nanoparticles for coating titanium implants associated with activation of ERK, inhibition of JNK, and enhanced BMP2 and BMP7 compared to control [71]. These results support that IGFBP3 overexpression diminishes osteoclastogenesis and enhances osteogenesis of titanium implants, and can serve as a potent molecule for increasing the efficacy of dental implants. Schleicher et al. reported that biomaterials constructs bound to IGF1 and IGFBP5 facilitate human osteoblast serum-free expansion in vitro, and enhance cell attachment, proliferation, and migration [72]. These results promote a potential application in surface modification of biomaterials for tissue reconstruction.

In the present paper, we focused on oral and facial cells with the aim of evaluating their site-specificities. Many studies were carried out on primary cultures of osteoblasts from rodent embryo calvaria as the gold standard of osteoblastic cells to decipher GH/IGFs/IGFBPs action but without any comparison with osteoblasts from other bone sites. However, the higher expression of IGF2 in mandibular osteoblastic cells than in iliac osteoblastic cells is in favor of an important and a site-specific role of the GH/IGF axis in oro-facial mineralizing cells [7]. The site-specific action of the GH/IGF axis was also supported by Shi et al., who compared the gene expression profiles of human DPSCs and BMSCs, as representative populations of odonto- and osteo-progenitors respectively, and showed a higher expression of IGF2 in DPSCs whereas IGFBP7 is more highly expressed in BMSCs [73]. Together, these results suggest a site-specific role of the GH/IGF axis in oro-facial mineralized tissues, where they could induce differential effects on cell proliferation, differentiation, survival with various consequences on bone, dentin, and enamel matrix synthesis tightly linked to tooth and oro-facial bone morphology.

## 5. GH/IGF Axis Modulates the Morphology of the Dento-Alveolar Complex and Cranio-Facial Bones

Craniofacial features associated with GH-related pathologies in humans and in transgenic animal models as well as the craniofacial effects of GH systemic treatments have been extensively reviewed by Litsas et al. in 2013 [4]. Yakar and colleagues exhaustively reviewed the effects and actions of the GH/IGF1 axis on body size, and the axial and appendicular skeleton, presenting human studies and animal models [6].

Most GH effects observed in human oral bone growth were reproduced by comparing cranio-facial morphology and dimensions in mouse transgenic models overexpressing GH or dwarf mice [74]. Among other defects, the length of both the upper and lower incisors and the angle of the mandible were significantly increased in the giant mice and reduced in the dwarf mice. This study showed that GH plays a major role in the growth and development of the craniofacial complex by directly and indirectly modulating the size and the angular relationships of the craniofacial structures, including the incisor teeth. Interestingly, IGF1 null mutant mice demonstrated a generalized decrease in craniofacial size and a non-allometric change of shape when compared with their wild-type littermates [75]. While the mandible did not exhibit any shape changes, the facial and cranial areas demonstrated prominent changes. Recently, Marchant et al. investigated the impact of genetic removal of IGF1 from endothelial cells using Wnt1-Cre; Vegfa^fl/fl^ mice [76]. This study showed that IGF1 is secreted by blood vessels and that conditional removal of IGF1 from blood vessels causes craniofacial defects including a shortened mandible. The authors concluded a crucial angiocrine role for IGF1 during craniofacial growth, and identified IGF1 as a putative therapeutic for jaw and/or cartilage growth disorders. Together, the GH and IGF1 transgenic mouse models support an effective action of GH and IGF1 in the mandible growth and shape.

Based on these clinical and experimental data, serum levels of IGF1 and GH have been shown to be relevant predictors of body size and skeletal acquisition (see [6] for review). Focusing on craniofacial tissues, it has been shown that circulating IGF1 levels correlate with cervical skeletal maturity [77], mandibular length [78], and total anterior facial height [79]. These observations led the authors to conclude that the IGF1 blood-spot test may be used as a biomarker for predicting the timing and intensity of mandibular growth as well as the height of the anterior face without the restrictions implied by radiographic techniques to assess skeletal maturity.

Craniofacial morphology and dental maturity in children with reduced somatic growth of different etiology have been recently reviewed [80,81]. Based on the published craniofacial and dental characteristics of children with isolated GH deficiency, with idiopathic short stature, small for gestational age and/or with intrauterine growth retardation, and of short-statured children of genetic origin (Silver-Russell syndrome, Turner syndrome, familial dwarfism), the authors concluded that children with short stature of different origins develop similar craniofacial characteristics, including smaller lengths of the cranial base and the mandible, and proportionately smaller posterior than anterior facial height, retrognathic face, and posterior rotation of the mandible. Of note, results on the length of the maxilla were contradictory. Dental maturation, however, does not demonstrate a specific pattern; the dentoalveolar anomalies involved slight retarded dental maturity and eruption, tooth crowding, anterior open-bite tendency, high incidence of distal bite and, for children with GH-deficiency and pituitary dwarfism, dental abnormalities such as microdontia [4,80,82]. Of note, the prevalence of dental decay for permanent and deciduous teeth does not significantly differ between children with GH deficiency and the control group [83]. Interestingly, although very rare, the GH-deficiency in patients with amelogenesis imperfecta (enamel defects due to mutations on enamel key genes) has also been reported [84,85].

In this context, the association between variants of the GHR coding gene and craniofacial tissue phenotypes has been exhaustively addressed [4]. However, since 2013, new GHR variants, such as P561T, rs6184 SNP and rs6180, have been identified as associated, in one hand, with the face mandibular morphology and tooth dimensions [2,9,86] and, in the other hand, with developmental defects of enamel [12]. Recently Marañón-Vásquez et al. showed that genetic polymorphisms in GHR (rs2973015) and IGF2R (rs2277071) have been associated with variations in craniofacial dimensions [87].

Overall, most of the data published on GH/IGF axis in teeth and oral-facial bones have shown the ability of cells to respond to GH and IGFs via their respective receptors, with a higher response in immature and proliferating cells than in differentiated cells. Excluding GH that seems not to be secreted locally, precise comparative patterns of expression of GHR and IGF system components in all cells of the dento-alveolar complex remain an open question. Discrepancies between the patterns of expression reported in the different studies may result from the various experimental models and methodological procedures and would need a systemic analysis of each element in each tissue of the dento-alveolar complex. One of the major features related to the high expression levels of GH/IGF components in the cells of the dento-alveolar complex is their effects on dental shape, structure, and eruption, as well as mandibular morphology. GH and IGFs, together or separately, are able to modulate the composition of enamel, dentin, and bone matrices by up-regulating proteins involved in these mineralizing processes, notably bone matrix proteins (BMP2 and BMP4), collagen 1α1, and EMPs with possible differential effects depending on the duration of the treatment due to feedback loops. In light of these observations, recombinant human GH (rhGH), rhIGF1, and rhIGFBPs, used topically or through systemic application, have been proposed for growth defects therapies, orthodontic treatments and regeneration of the dento-alveolar complex.

## 6. Therapeutic Systemic Applications of GH or IGF1 for Craniofacial Tissue Defects

### 6.1. GH Systemic Therapy in Patients with Growth Defects

The stimulation of growth by injection of rhGH in children with hypopituitarism was first reported more than 45 years ago. Since the approval of rhGH treatment by the Food and Drug Administration and other health agencies, some short children with or without GH deficiency, such as children treated with total body irradiation and bone marrow transplantation [88,89], are now being selected for GH substitution therapy. Treatment of children with rhGH produces a “catch up” phenomenon in both height and skeletal maturation, regardless of their diagnostic differences.

Craniofacial measurements in GH-treated children are limited but the published results suggest that after GH administration, the exact amount and pattern of growth is unpredictable; however, the facial convexity decreases, whereas mandibular length and posterior facial height increase. The data on the impact of GH treatment on dental morphology and maturity remains controversial. Indeed, based on the literature, Davidopoulou and Chatzigianni conclude that tooth eruption remains unaffected after GH therapy [80], while several studies suggest a decrease in the delay in tooth eruption, a significant acceleration of dental age and reduction of disparities in the size of jaws and teeth after applying hormonal therapy in short stature children [82,90,91].

### 6.2. GH and IGF Systemic Treatments in Animal Models

The effect of GH systemic therapy on odontogenesis was examined in Lewis (control), dwarf (Dw), and Dwarf GH-treated (Dw + GH) rats, which received twice-daily rhGH dose of 65 mg/kg from the post-natal day (PND) 2 to PND15. This study showed that rGH has an effect in enamel mineralization and root development. In 6-day-old rats, enamel mineralization is delayed in Dw and Dw + GH animals. Similarly, the root initiation is also delayed to 9 days in Dw and Dw + GH rats compared to 6 days in controls [92]. In another study, rats treated twice-daily with rhGH (1 mg/kg) for 5 days, the molar odontoblasts lining the pulp, cementoblasts, osteoblasts, and PDLCs showed increased expression of differentiation markers, BMP2, BMP4, ALP, OCN, and OPN, and increased number of these cells [48,93]. Together, these results show that GH influences rat dentinogenesis and root formation during dental development.

In a rat model of acromegaly in which a prominent mandibular enlargement was induced by a 4-week continuous subcutaneous infusion of rhIGF1 (640 µg/day) in adult rats, the length of the mandible, maxilla, and femur is increased. Contrary to other bones, the rat mandible does not return to control size after IGF1 treatments and the disharmonious jaw size (between maxilla and mandible) persisted even after circulating IGF1 levels normalized. These findings, showing a bone site-specificity for GH/IGF1 action, suggest that mandibular occlusal treatment should only be considered for acromegalic patients when serum IGF1 levels are normalized and bone growth ceased [94].

## 7. GH/IGF and Oral Tissue Engineering

### 7.1. Dentin Lesions, Reparative Dentinogenesis, and Pulp-Capping Materials

The dentin–pulp complex of the tooth has inherent natural regenerative properties. In fact, following deep dentinal lesions, odontoblasts lining the pulp chamber are altered and DPSCs are recruited. They proliferate and differentiate into odontoblast-like cells in order to secrete a tertiary dentin. However, this spontaneous regenerative property is limited in very deep dentinal lesions close to the pulp with/without pulp exposure and vital pulp therapy (VPT) is often required to maintain pulp vitality (see [95] for review). VPT includes removal of infected dental tissues and pulp capping. This procedure involves two treatments options: either the capping agents are placed directly over the exposed pulp (direct pulp capping), or a cavity liner or sealer is placed over a residual thin layer of affected and non-infected dentin that covers pulp chamber (indirect pulp capping) and, subsequently, a permanent restoration is placed using restorative materials (Figure 3A).

Pulp capping requires capping materials (e.g., calcium hydroxide, mineral trioxide aggregate (MTA), zinc oxide eugenol cement) that can be used alone or combined with bioactive molecules to stimulate DPSCs recruitment and differentiation. Interestingly, pulp capping materials alone have been shown to solubilize bioactive dentin molecules that stimulate tertiary dentinogenesis. The non-collagenous proteins present in the dentin extracellular matrix (ECM) include matrix molecules (DSP, DPP, BSP, DMP1, and DSPP), both anti- and pro-inflammatory chemokines and cytokines (TNFα, IL1, IL6, and IL10) and growth factors such as TGFβ1, BMP7, NGF, GDNF, and bFGF, but also IGF1 and IGF2 (see [95] for review). To characterize the growth factors released from dentin by pulp-capping agents, Tomson et al. exposed over 14 days powdered human dentin to calcium hydroxide and white and grey MTA and analyzed the solubilized components using multiplex quantitative ELISA [96]. Interestingly, their results showed that a broad range of bioactive molecules which induce in vitro proliferation and chemotaxis of primary rat DPCs, are released from dentin by the pulp-capping agents. Among these released dentin ECM components, IGFBP1 was detected.

In addition to this materials-based approach, some authors have also proposed the use of progenitor cells such as SCAPs, with or without scaffolds, whose properties may be also used in emerging regenerative endodontic procedures (REPs) as a treatment option for immature necrotic teeth to allow the reestablishment of a newly formed vital tissue and enable continued root development [21,23,97]. Indeed, as previously mentioned in Part 4, SCAP proliferation, migration, and formation of mineralized nodules is stimulated by IGF1 in a mixture of growth factors naturally present [65].

Other studies have also demonstrated variations of the GH/IGF components in DPCs response to dentin lesion. In 2005, McLachlan et al. showed that GHR expression decreases in carious pulpal tissue compared to healthy pulp [98]. In 2006, Masuda et al. showed that laser irradiation of rat incisors enhances GHR and IGFBP5 expression in addition to other genes related to odontoblast-differentiation and formation [99]. In 2018, Alkharobi’s group also demonstrated a novel role of IGFBP2 and IGFBP3 in cell response to mild inflammatory environment associated with superficial caries by showing that DPCs from teeth with superficial caries display higher levels of IGFBP2 and IGFBP3 when compared to cells isolated from healthy teeth [100].

Together, these data highlight the involvement of the GH/IGF components in dentin repair and have motivated the use of recombinant proteins in VPTs. Indeed, several studies aimed to evaluate the action of rhGH and rhIGF1 in dentin repair. In deep class V dentin lesions with cervical lesions located on the buccal surfaces of the maxillary and mandibular permanent teeth in dogs, bioactive factors (IGF1, BMP7, bFGF, TGFβ1) were applied to the prepared cavities [101]. The authors found that tertiary dentinogenesis and intratubular mineralization are not different in the IGF1-treated group, but greater in the TGFβ1-treated teeth compared to the other groups. However, two other studies carried out in rats [102] and rabbits [103] with dentin lesions with pulpal exposure, reported a beneficial effect of IGF1 when the dentin-pulp capping was performed with IGF1.

### 7.2. Dental Movements Induced by Orthodontic Treatments

The dental movements induced by orthodontic treatments are tightly dependent on periodontal tissue and alveolar bone remodeling ability, associated with both osteoclast and odontoblast activities. The forces delivered by orthodontic appliances generate compression and traction zones on both sides of the dental root. A gap in the alveolar resorption and/or the dental root appears in the compression zone and a bone apposition phenomenon occurs in the traction zone.

#### 7.2.1. GH/IGF Axis and Dental Movement

Orthodontic tooth movements appeared to up-regulate GHR and IGF-1R immunoreactivity [104]. The number of IGF1-, IGF-1R-, and IRS1-positive cells increase significantly on the tension side and decrease on the compression side. These data indicate a close relationship between the mechanical loading of the PDL and the autocrine/paracrine expression of the components of the IGF system as an early step in the mechanotransduction process leading in the long term to an organized remodeling of the alveolar bone. Orthodontic forces applied for 9 days on the maxillary first molar in rats increase root resorption with an increase in IGF system components, including IGF-1R, IGF2, IGFBP5, and IGFBP6, observed in pressure areas and resorption gaps [42]. These results suggest an involvement of the IGF system in the resorption-repair sequence, which is a known bone coupling process. Conversely, IGF1, IGF-1R, and PCNA (Proliferating Cell Nuclear Antigen) are less expressed in PDL cells of hypofunctional groups than in controls. These results suggest that occlusal stimuli via orthodontic tooth movements induce cell proliferation of PDL cells by increasing IGF1 and IGF-1R expression [105]. Interestingly, this overexpression of IGF1 may be enhanced in maxillary alveolar bone when occlusal stimuli are associated with physical activity [106]. Of note, recent studies also showed that effects of intermittent parathyroid hormone (PTH) on the stability of orthodontic retention are improved by IGF1 [107].

Bone remodeling necessary for orthodontic tooth movements involves active osteoclasts, which are positive for tartrate-resistant acid phosphatase (TRAP) activity and which may also be regulated by GH via GHR. Orthodontic tooth movements appeared to up-regulate GHR expression along rat alveolar bone, root surfaces, and PDL [104,107]. TRAP and GHR-positive cells were increased in the compression side along the alveolar bone, root surface, and in the PDL space in rats treated with orthodontic appliances for tooth movements [108].

#### 7.2.2. GH and Dental Movement

GH effects were evaluated on PDL and alveolar bone during dental movements in rats receiving 30 N orthodontic forces on the maxillary first molar [109]. Collagen fibers and the number of osteoclasts in the compression zone are significantly increased while the number of blood vessels is decreased. These results suggest that GH promotes the acceleration and intensification of bone resorption and the formation of immature collagen fibers during orthodontic treatment.

In another study, Hu et al. induced mesial movement of the rat maxillary first molar by applying an orthodontic force of 50 g. A significant increase in the number of osteoclasts and RANKL was observed 3 days after application of the force. RANKL is the main factor involved in the differentiation and activity of osteoclasts, whose activity is inhibited by OPG. The situation was different after 7 and 14 days of treatment, when a decrease in the resorption index and in osteoclasts was observed with an increase in the RANKL/OPG ratio. These data allow to conclude that despite the early alveolar resorption, which confers an activating effect to GH, GH finally inhibits root resorption by decreasing the RANKL/OPG ratio and up-regulating the expression of IGF1 [110].

### 7.3. Periodontal Regeneration

The therapeutic effects of GH and IGF1 and, more precisely, the biological actions of PDGF, TGFβ1, FGF, IGF1, and EGF on periodontal cells and tissues have been extensively investigated (Figure 3B) (see [111] for review). We provide here an update of the data on rhIGF1.

#### 7.3.1. IGF1 and Periodontal Regeneration

In 1996, Giannobile et al. characterized the biological effects of growth factors used individually or in combination on periodontal regeneration. In this comparative study, periodontal disease was induced by ligation with pocket depth greater than 6 mm and alveolar bone loss. After periodontal surgery, methylcellulose gel containing or not 10 µg PDGF-BB or 10 µg IGF1 was applied, alone or in combination to the exposed root surfaces. Application of IGF1 alone did not significantly improve healing but, when combined with PDGF-BB, it increased the attachment formation and bone filling [112]. Five other studies reporting the effects of the local application of PDGF/IGF1 combination on periodontal regeneration were published between 1989 and 2014 [93,94,95,96,97]. Combination of PDGF and IGF1 improves periodontal regeneration in dogs with periodontal disease. The affected sites treated with the growth factors had significant amounts of new formed bone and cementum [113]. In 1991, the same team replicated the study using conventional periodontal surgery (flap reflection, debridement, and root planing). PDGF/IGF1 combination improves the cementum and alveolar bone neoformation [114]. In alveolar bone defects in female dogs filled with calcium phosphate-based biomaterials loaded or not with the PDGF/IGF1 combination, bone formation was more important when the biomaterials were loaded with the growth factor combination [115]. Finally, in order to evaluate the potency for bone formation of IGF1 alone or combined with PDGF, associated with liposomes, Abreu et al. extracted the maxillary second molar in rats. Histological analysis showed that alveolar healing was more pronounced in the groups treated with isolated or combined growth factors in liposomes than liposomes alone [116]. In 2005, Soares et al. reported observations in contradiction with the other studies [117]. The repair of experimental class II furcation lesions in dogs was analyzed after application of a tissue graft derived from extraction sockets previously treated with a combination of PDGF and IGF1 diluted in methylcellulose gel. The grafts were harvested from the alveoli of the upper 2nd and 3rd premolars and inserted into the lesions induced in the mandibular 2nd, 3rd, and 4th premolars. The authors could not detect any significant differences between the treated sites and the control sites in the measured parameters such as junctional epithelium, neoformed alveolar bone, and neoformed cementum, suggesting that neither IGF1 nor PDGF present any therapeutic advantage in this model.

Other studies have been conducted with IGF1 encapsulated in biomaterials. In an experimental model of class III furcation defect in dogs, IGF1-loaded in dextran-co-gel hydrogel microspheres promotes a significant increase in neoformed bone in periodontal defects compared to IGF1 in blood clot [118]. In 2012, Zairi et al. compared the inflammatory and tissue responses to IGF1 and three other growth factors in perforations of the pulp floor and endodontic treatment in dogs. The authors observed higher rates of epithelial proliferation in the TGFβ1, bFGF, and IGF1 groups compared with mineral trioxide aggregate, and higher rates of cementum formation in the IGF1 and bFGF groups compared with zinc-oxide-eugenol based cement [119].

Two other studies reported the effects of a systemic administration of rhIGF1 on alveolar bone in healthy and diabetic rats. In the first one, healthy rats had an avulsion of the right mandibular first molar and received a subcutaneous infusion of 320 mg/day of IGF1 for 3 weeks. Micro-computed tomography (microCT) analysis showed an increase in neoformed alveolar bone during and after IGF1 administration and a significant reduction in alveolar ridge height loss after dental avulsion [120]. In diabetic rats, the systemic treatment with IGF1 also improved the healing of the alveolar bone after dental avulsion [121].

Together, these encouraging results have led to the implementation of clinical trials. In humans, phase I and phase II clinical trials have shown that the efficacy of topical application of IGF1 in combination with PDGF in periodontal healing is dose dependent. At high doses (150 µg of each), the periodontal healing, characterized by an increase in the height of the alveolar bone as well as by an optimal filling of the bone defects, was very satisfactory [122]. In another randomized controlled clinical trial including patients with periodontal disease, 10 µg rhIGF1 combined with 0.8 µg rhVEGF resulted in better bone healing than when these factors were used individually [123].

#### 7.3.2. IGF1 and Sockets in Medication-Related Osteonecrosis of the Jaw (MRONJ)

The mandibular bone of patients with certain pharmacological treatments against osteoporosis may be subject to osteonecrosis after tooth extraction [124,125]. The mechanisms involved in this pathological process are not yet established and relate jaw bones exclusively. In MRONJ model generated in Wistar/ST rats by zoledronate and dexamethasone subcutaneous injections for 2 weeks and unilateral maxillary molars extractions, a mixture of IGF1 and other cytokines improved the healing with complete soft tissue coverage and socket bones, whereas in the other groups, the exposed necrotic bone with inflamed soft tissue remained unrepaired [126]. Histological analysis and immunohistochemical staining against CD31 showed new bone formation, the presence of osteoclasts, and an increase in endothelial cell number in the cytokine mixture. These data showed complete mucosal healing, bone regeneration, and angiogenesis in the extracted alveolar sockets suggesting that intravenous administration of IGF1 in cytokine mixtures might be an effective therapeutic approach for treating jaw osteonecrosis.

### 7.4. Osseointegration of Oral Implants

Changes in surgical procedures, topography and surface chemistry of biomaterials have led to significant improvements in implant osseointegration, initially described by a direct structural and functional connection between the surface of a load-carrying implant and the ordered, living crestal bone (see [127] for review). Although most of the attention has been focused on the surface preparation of the titanium implant, many projects have proposed to improve and accelerate osseous healing using topical treatments (Figure 3C). Most of them include the application of platelet-rich plasma (PRP), BMPs, as well as IGF1, IGFBPs, and GH.

#### 7.4.1. IGF1 and IGFBP3 and Osseointegration

To our knowledge, no study has so far investigated the effect of topical application of IGF1 alone on implant osseointegration within the residual tooth socket, only the combinations of IGF1 with PDGF or IGF1 with VEGF have been used. The effects of topical administration of IGF1 alone have only been investigated on the sealing of the peri-implant epithelium around the implant (Figure 3C).

Three studies have investigated the combined effects of IGF1 and PDGF on the peri-implant bone healing in dogs. In 1991, Lynch and colleagues evaluated the early wound healing events of bone around dental implants [128]. Mandibular premolars of beagle dogs were extracted and submitted or not to a local treatment with PDGF/IGF1 and nine months later, the extracted teeth were replaced with an implant. Histometric evaluation performed at 1 and 3 weeks after implant placement revealed that the percentage of bone fill in the peri-implant spaces and the percentage of implant surface in contact with the new bone are significantly increased in PDGF/IGF1 treated sites. Based on these results, the authors suggested that the PDGF/IGF1 mixture can stimulate bone regeneration around titanium implants.

Unlike this experimental procedure where PDGF/IGF1 and the implant were placed in the socket after a long period of healing, the effect of topical application of PDGF/IGF1 on the osseointegration of dental implants placed immediately in fresh extraction sockets has been investigated in two different studies [129]. When the mandibular premolars were removed in mongrel dogs, with an application of 5 µg/mL of PDGF and IGF1 and the insertion of implants, histometric analysis showed a greater extension of the bone/implant contact in the treated groups than in the controls, but without significant differences in intensity of bone labeling (using calcein administration). The authors concluded that the combination of PDGF/IGF1 actively takes part in the initial phase of bone repair. A similar experimental procedure was used by the same team except that the extraction sockets were widened before implant placement in order to mimic the clinical situation of immediate implants placement in humans, wherein frequently the implant is partially in contact with the bone host [130]. The histometric results at 3 and 8 weeks after surgery show greater extension of bone-to-implant contact, larger percentage of bone area, and greater intensity of bone staining for PDGF/IGF1 treated than control implants. Based on these data, it was concluded that the combination of PDGF/IGF1 might be a good alternative for enhancing bone healing around implants partially in contact with bone.

More recently, the effects of topical administration of rhIGF1 were investigated on the sealing of the peri-implant epithelium around the implant [131]. Maxillary first molars were extracted from Wistar rats and immediately replaced with implants. Three weeks post-implantation, cotton floss was immersed in 50 ng/mL IGF1 solution and laid onto the gingival margin around the implant body every 2 days for 2 weeks. Histomorphometric analysis showed that the implant-PIE interface exhibited a band of immunoreactive laminin-332, similar to the tooth-junctional epithelium interface, that was partially absent in the untreated group. These data suggest that rhIGF1 is effective in improving epithelial integration around titanium implants. Interestingly, IGF1-treated implants were shown to generate more mineralized tissues, and presented stronger expression of RUNX2, SP7, and OCN than the control group [36].

The osteogenic efficacy of chitosan gold nanoparticles (Ch-GNPs) conjugated with IGFBP3-coated titanium implants was evaluated on Sprague Dawley rats with extraction of the lower first molar. One month after the extraction, the implant site was prepared using a 0.8 mm-diameter drill and Ch-GNP/IGFBP3-coated implants were placed in the extraction socket. Four weeks post-implantation, microCT analysis demonstrated that IGFBP3 increased the volume of newly formed bone surrounding the implants with decreased level of the osteoclastogenic molecule, RANKL, and enhanced expression of osteogenic molecules including BMP2, BMP7, and OPN. The authors concluded that IGFBP3 overexpression diminishes osteoclastogenesis and enhances osteogenesis of titanium implants, and that IGFBP3 might serve as a potent molecule for osseointegration [71].

#### 7.4.2. GH and Osseointegration

GH supplementation on osseointegration has been exhaustively reviewed (see [132] for review). To date, only five studies have investigated the effect of GH on implant osseointegration in orofacial bone sites.

In 1997, in order to model the effect of systemic administration of GH, titanium implants were placed in the forehead (nasal cavity) of transgenic mice overexpressing bovine GH (bGH) [133]. The mean levels of bGH in the transgenic mice were 1124 ng/mL, on average ten times higher than the peak levels of mouse-GH found in normal mice. Histomorphometric analyses performed 4 months after surgery demonstrate significantly more direct bone-to-metal contact in transgenic mice than in wild-type littermates, suggesting that systemic administration of GH in humans may improve the integration of the implant into the bone.

The effect of topical application of GH on the osseointegration of dental implants after a healing period was investigated in two studies conducted by the same teams [134,135]. In their experimental procedure, maxillary and mandibular premolars and molars were extracted from beagle dogs. After two months of healing, each mandible received implants with prior application (or not) of 4 IU (1.2 mg) rhGH. After a 2-week treatment period, when compared to control implants, the GH-treated sites showed an increase in the perimeter of bone in direct contact with the treated implants, total peri-implant area, and new bone formation, whereas no significant increase in inter-thread bone in GH-treated implants are observed [134]. After 5 and 8 weeks, topical application of GH during implant placement had no significant effects on the bone-to-implant contact, although bone neoformation and inter-thread bone values increased significantly [135]. The topical application of GH may thus act as a bone stimulant in the placement of endosseous dental implants. However, further studies are necessary to clarify these results and assess newly formed bone. Similar procedure using 4 IU (1.2 mg) rhGH combined with melatonin (1.2 mg) showed that GH and melatonin synergistically enhanced new bone formation around titanium implants in early stages of healing [136].

Studies investigating the effects of topical application of GH around immediate dental implants in fresh extraction sockets of mongrel dogs showed a higher bone density and more well-oriented collagen fibers in the GH-treated group by the 12th week after the implantation. Here, right and left first premolars were extracted and powdered rhGH (4 IU—1.2 mg) was immediately placed in the socket, and thereafter, the implant immediately placed (left side), while implants were placed without GH at the control site (right side). The authors concluded that the use of GH powder around dental implants placed immediately in fresh extraction sockets enhanced peri-implant bone response [137].

### 7.5. Craniofacial Bone Repair

#### 7.5.1. GH/IGF Axis in Distraction Osteogenesis and Bone Healing

In parallel to their application for the optimization of implant osseointegration, the use of GH/IGF molecules for orofacial bone repair has also been investigated. The healing of mandibular and cranial fractures, defects, or osteotomies is a clinical challenge in reconstructive surgery that may be useful for patients with deregulated GH. In particular, distraction osteogenesis is a bone-regenerative process in which an osteotomy is followed by distraction of the surrounding vascularized bone segments, with formation of new bone within the distraction gap (Figure 4). This surgical procedure has proven successful in treating a variety of patients with craniofacial abnormalities such as mandibular growth defects (see [138] for review). The development of innovative tissue engineering strategies including the use of bioactive recombinant proteins may considerably improve the bone healing.

Elevated levels of IGF1 in the serum and bone tissue were observed in the first phases of healing in a fractured jaw healing rat model [139] and during the healing of calvaria in a critical defect rat model [140]. Likewise, in addition to BMP2, BMP4, TGFβ1, and bFGF, both IGF2 and IGF1 were shown to be induced/upregulated during distraction in various cells located in the distracting region of the mandible [141,142,143,144]. During distraction, IGF1 is thought to be responsible for osteoblast proliferation and osteoblast differentiation from precursor mesenchymal cells, leading to de novo bone formation [141] and has been shown to mediate the osteogenic beneficial effect of mesenchymal stem cells transplantation into distracted callus [145]. During distraction osteogenesis in limbs, in addition to a systemic increase in bFGF, TGFβ1, IGF1, IGFBP3, and GH in serum [146], GHR expression was also increased, suggesting an enhanced sensitivity to endogenous systemic GH and thus the promotion of consolidation of the regenerated bone [147]. Together, these studies support that the use of IGF1 and GH alone or combined with other bioactive recombinant proteins and/or biomaterials may be of benefit during distraction osteogenesis and orofacial bone healing.

#### 7.5.2. IGF1 and GH in Distraction Osteogenesis

In 1999, Stewart et al. conducted a study to determine the effects of local rhIGF1 during mandibular distraction using a rabbit model of osteotomy and distraction lengthening [148]. Ten mg of rhIGF1 were administered using osmotic infusion pumps over 28 days directly into the distraction gap of each hemi-mandible. Four weeks after distraction, while DEXA scanning and three-point bending failed to detect any effect of distraction rate of IGF1 infusion, histological and histomorphometric analysis revealed that IGF1 infusion significantly enhances osteoblastic activity indicating that exogenous IGF1 has a positive influence on osteoblastic activity during distraction.

GH effects were investigated on early bony consolidation in mandibular distraction osteogenesis after a vertical osteotomy at a rate of 1 mm/d up to a 10-mm distraction [149,150]. When daily subcutaneous injection of rhGH (100 µg (1 IU) per kg of body weight) was performed from the day of the osteotomy through the 10-day-distraction period, the level of serum IGF1 was elevated and peaked after 8 days and 12 days. Two, four, and six weeks after completion of distraction, bone mineral density and bone mechanical strength were higher in the rhGH group when compared to the control one. Histological examination showed the formation of a substantial amount of active woven bone throughout the distracted zone at 6 weeks in the rhGH group, whereas, in the control group, most of the central area of the distracted zone was filled with fibrous tissue [149].

The second study aimed at studying the effect of chitosan microsphere-encapsulated rhGH implantation (4IU) in the same distraction model [150]. Bone mineral density, three-point failure load, and histological analysis performed 3 and 6 weeks after completion of distraction led the authors to suggest that chitosan microsphere-encapsulated hGH seems to be effective in early bone consolidation in distraction osteogenesis [150].

#### 7.5.3. IGF1 and Mandibular Bone Defect

In 2003, Deppe and colleagues investigated the effect of a combination of titanium membranes coated with rhIGF1 and rhTGFß1 (1% *w*/*w*) in the regeneration of transosseous defects created in the mandibular rami of Sprague-Dawley rats [151]. Clinical observation and histometric analysis performed after a healing period of 28 days led the authors to conclude that a combination of TGFß1 and IGF1 does not improve bone quality. Nevertheless, the coating titanium membranes with the mixture of both growth factors, TGFß1 and IGF1, can accelerate the healing process of bony defects in the rat mandibular model. Similarly, IGF1 (25 ng) and TGFß1 (0.1 µg) incorporated into a hydrogel scaffold induced bone regeneration in defects created in mandibular ramus of Sprague–Dawley rats [152]. Radiological and morphological analysis showed a significant bone formation after 6 weeks, leading the authors to conclude that the hydrogel scaffold impregnated with IGF1 and TGFß1 can induce bone regeneration and is therefore a potential surgical tool for the enhancement of surgical repair of bone defects.

#### 7.5.4. GH and Calvaria Bone Defect

In 2001, Cacciafesta et al. investigated the healing effect of a systemic administration of GH (2.7 mg/kg/day for 28 days) on calvarial bone defects surgically created on parietal bones of Wistar rats and covered with expanded polytetrafluoroethylene (e-PTFE) membranes [153]. Three weeks after surgery, mechanical testing and microCT analysis revealed that ultimate load, ultimate stiffness, energy absorption at ultimate load, and tissue dry and ash weights, together with bone volume were significantly augmented in the rhGH-treated group compared with the placebo group. The experiment demonstrated that rhGH administration enhances bone deposition and mechanical strength of healing rat calvarial defects, covered with e-PTFE membranes. In 2009, Tonietto et al. also conducted an in vivo study in Wistar rats to evaluate the effects of the incorporation of rhGH (4 IU) and fibers of PLGA/poly (isoprene) blend in a scaffold of α-TCP inserted calvaria bone defects [154]. After a 2-week period, the groups with biomaterials charged with rhGH were significantly superior with regard to angiogenesis and bone neoformation. In the 3-week period, the group with α-TCP charged with rhGH was significantly superior to the other groups with regard to bone neoformation. These results suggest that α-TCP scaffold may have improved osteoconductive capacity when rhGH is incorporated into its structure. In contrast to these results, Chaves et al. recently showed that the utilization of autogenous bone grafts associated with local application and irrigation with 0.4 IU rhGH does not improve the bone repair in calvarial bone defects in Wistar rats [155]. Notably, many kidney alterations were observed in treated rats. Based on the renal and hepatic histological results, the authors concluded that the intense resorption of autogenous bone graft particles induced by the GH catabolic activity is the major factor of the renal alterations and warn of potential hepatic and renal side effects promoted by locally-delivered GH.

## 8. Discussion

GHR and IGF-1R are present in most, if not all, mineralizing cells of the dento-alveolar complex, with a higher expression in immature proliferating cells and stem cells than in full differentiated ones, and with a higher expression in oral bone cells than in appendicular bones when systematically compared. The high expression levels of GH and of the IGF system components in oral mineralizing cells suggest an important role of GH/IGF axis in dental tissues and oro-facial bones.

In cells producing collagen 1α1 for mineralization such as osteoblasts and odontoblasts, GH induces expression of osteogenic markers, BMP2 and BMP4 among others, which play an active role in cell differentiation and bone matrix mineralization. In ameloblasts that do not produce collagen 1α1, the putative role of GH is less clear as amelogenesis mainly occurs before birth when GH is not yet present. However, enamel defects in some patients with GHR variants or mutations suggest a role of GH/GHR in enamel synthesis that needs to be investigated. In dental cells, GHR, IGF-1R, and IGFBPs that modulate IGF actions, are very dependent on the cell types considered and on their stages of differentiation [25,26,27,28,29,30,31,32,33]. Thus, IGF-1R expression is high in proliferating immature pre-ameloblasts, decreases in secretion-stage ameloblasts, which secrete enamel matrix proteins, is particularly low during the transition stage when around 25% of cells are lost by apoptosis, and increases again in maturation-stage ameloblasts in charge of terminal enamel mineralization with particularly high expression in ruffle-ended ameloblasts that actively control calcium and proton transport for optimal enamel mineralization. IGF-1R expression finally decreases at the end of amelogenesis, before eruption, when ameloblasts apoptosis is again elevated, suggesting a determinant role of IGFs in ameloblasts differentiation and function that need to be precisely investigated. IGFBP expression is also modulated during amelogenesis with IGFBP2 and IGFBP3 mainly expressed during the maturation stage, and with IGFBP5 and IGFBP4 to a lesser extent [31,156] (our unpublished data). Interestingly, IGFBP6 seems essentially present in Hertwig epithelial root sheath cells, which are the remaining epithelial cells in the radical part of the teeth where it likely limits their proliferation induced by locally produced IGF2. The discrepancies reported in the literature concerning their relative level of expression in the different oral cells, and the presence of IGFBP1 are probably due to technical bias and a systemic analysis would help to answer this question.

In odontoblasts, GHR and IGF-1R, as well as IGF2 and IGF1, are expressed all along the dentinogenesis, with a high expression in early differentiation stages concomitant with that of IGFBP2 and IGFBP3 and, a continually decreasing expression during the whole differentiation process with IGFBP5. Although, the role of GH in dentinogenesis is clearly established by clinical observation of teeth in patients as well as in transgenic mouse models [34], its mechanism of action is less clear and questions remain regarding its local production, its direct and indirect effects on odontoblasts via GHR and IGF-1R, respectively, as well as the precise IGFBP actions that were poorly investigated in transgenic models. IGFBP5, mainly produced in mature odontoblasts, certainly plays a major regulatory role in dentin mineralization [156], whereas IGFBP3 seems more implicated in earlier stages and pre-dentin deposition [37,38]. GH, IGF1, and IGFBP5 are also involved in the physiological dentin repair forming the tertiary dentin [99]. rhIGF1 was shown to improve dentin repair in experimental dentin lesions in rats [102] and rabbits [103], but not in dogs [101]. IGF1 differentiating effects reported in dental pulp cells differentiation (DPCs and DPSCs) and SCAPs [66,68] make it a bioactive molecule worthy of interest for future innovative therapeutics for dentin lesion repair in adult or forming teeth, therefore allowing the tooth preservation instead of its removal in various pathological situations, including tooth decay, fracture, or infection.

Among the most promising applications of GH/IGFs/IGFBPs in oral tissue engineering, are orthodontic therapies, periodontal regeneration, implant osseointegration, and distraction osteogenesis, all tightly associated with bone remodeling. GHR and IGF-1R are highly expressed in oral bone cells, in osteoblasts and osteocytes with GH increasing major bone markers (BMPs, collagen 1α1, OCN, ALP, DMP1), and IGF1, also able to modulate expression of these markers as well as expression of the bone master gene, RUNX2. IGFBP5 promotes the osteogenic and the odontogenic actions of IGF1, whereas IGFBP2 seems to inhibit them. IGFBPs, including IGFBP2, IGFBP5, as well as IGFBP3, are released from the bone matrix during tissue remodeling, thus possibly modulating IGF effects and exerting their own independent effects notably on cell survival. Importantly, IGFBP6 which is expressed in HERS and in human BMSC and which has the highest affinity for IGF2 with cell protecting properties [157,158], may be also considered as an interesting future therapeutic approach in oral tissues. Recombinant GH, IGF1, and IGFBP3 have been successfully tested for therapeutic applications for implant osseointegration and bone regeneration, alone or associated with materials, where they showed their capacity to improve bone modeling.

Although these strategies are very promising, experimental studies are still needed to better understand the molecular mechanisms involved in this cell repair, as well as the long-term effects of these materials and their reproducibility. Indeed, in most studies presented in this review, bone formation was only investigated using histomorphometric analyses based on histological sections, over short periods of follow-up, with no dose-effects studies, and mainly with topical treatments and no quantification of bioactive factors in the bone site and no checking of the long-term safety of such treatments.

Considering the plethoric data on recombinant GH/IGF in bone/cartilage maintenance and repair leading to the development of rhIGF1 as a therapeutic and preventive strategy against osteoarthritis, considering the continuous innovations in hydrogels and scaffolds used in bone repair (see [159] for review), considering the considerable progress in macromolecule synthesis and the multiple strategies of their delivery, including adenovirus, lentivirus, and now direct RNAs, the present review overviewed the recent knowledge on the GH/IGF axis with a focus on oral stem and mineralizing cells, thus allowing the opening of many perspectives to set up innovative strategies for the repair of the dento-alveolar tissues. These strategies include combinations of recombinant proteins associated or not with biomaterials or nanoparticles, and associated or not with the highly multipotent oral stem cells.

## 9. Conclusions

The GH/IGF axis is not only crucial for the development of the dento-alveolar complex, but also for oral bone, ligament, and dentin repair. The vast majority of studies investigating the effects of the GH/IGFs/IGFBPs concluded on encouraging data on bone healing following implant placement and distraction osteogenesis in various experimental animal models (mice, rats, and dogs). Combinations of GH/IGFs/IGFBPs with other molecules such as TGFβ1, PDGF, or the use of PRP, an autologous product that contains highly concentrated platelets with growth factors seem promising therapeutic approaches that would enhance the healing of bone defects and counteract bone resorption. Combination of bioactive molecules with oral stem cells, which is easily accessible and presents great potentiality in tissue healing [160], could be a promising strategy for targeted oral tissue therapies with local treatments, associated or not in materials, for orthodontics, and the repair and regeneration of the dento-alveolar complex and oral bone.

## Figures and Tables

**Figure 1 cells-10-01181-f001:**
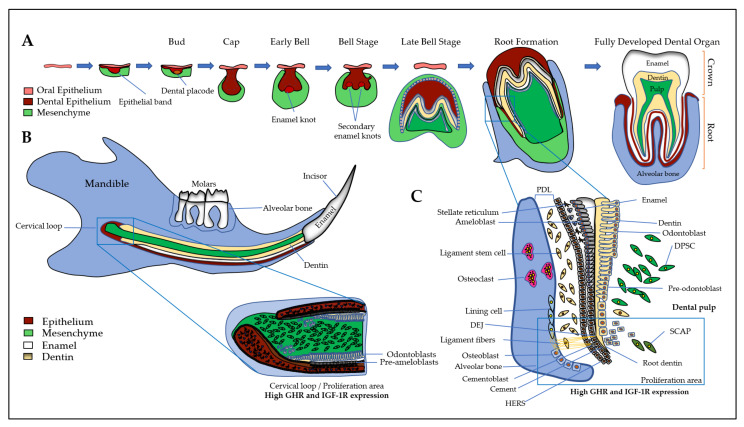
The dento-alveolar development. (**A**) Epithelial cells (in brown) interact with ecto-mesenchymal cells (in green) forming the bud, cap, and bell. The final erupted tooth is formed with the crown and the root anchored in the alveolar bone. The most external layer of the crown is the enamel (in white), which is synthesized by ameloblasts. The dentin (in yellow) is synthesized by the odontoblasts and the pulp contains DPCs, DPSCs, nerves, and vessels; (**B**) The rodent continually growing incisor summarizes the whole process of odontogenesis. First, precursor cells (expressing high levels of GHR and IGF-1R) are proliferating in the cervical loop, then cells differentiate either into ameloblasts that secrete enamel matrix proteins (in white), or into odontoblasts that synthesize the dentin (in yellow). At the end of amelogenesis, ameloblasts are lost, leading to an acellular and irreparable enamel matrix; (**C**) In human and rodent molars, the precursor cells are localized near the forming root involving odontoblasts, HERS, osteoblasts, and osteoclasts of the alveolar bone. During enamel synthesis, ameloblasts differentiate and change their shape and function. During dentin synthesis, odontoblast body cells move away from the DEJ, thus reducing the volume of the pulp chamber. The space between the dental root and the alveolar bone is formed by the fibroblasts, PDLCs and the cementoblasts lining the tooth, forming ligament fibers that attach the tooth to the bone. SCAPs are associated to the apex of a developing root, they may be recruited in case of necrotic pulp in order to complete root development and apexogenesis. DEJ: dentin- enamel junction, DPCs: dental pulp cells, DPSCs: dental pulp stem cells, GHR: growth hormone receptor, HERS: Hertwig epithelial root sheath, IGF-1R: insulin-like growth factor 1 (IGF1) receptor, PDLCs: periodontal ligament cells, SCAPs: Stem cells from the apical papilla.

**Figure 2 cells-10-01181-f002:**
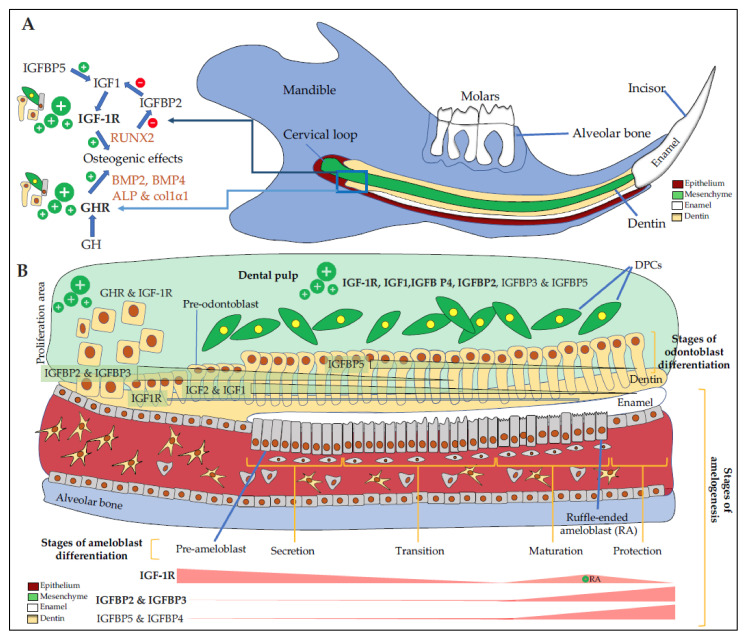
Expression of IGF components in the dento-alveolar complex. (**A**) Schematic representation of a sagittal section of a rodent hemi-mandible with the bone in blue, the teeth (3 molars and the growing incisor in white), the dental epithelium in red, the dental mesenchyme in green, and the dentin in yellow. There are many immature proliferating cells in the selected area especially in the cervical loop with cells expressing high levels of GHR and IGF-1R. Osteoblasts, odontoblasts, cementoblasts, periodontal ligament cells are responsive to GH, which increases the level of differentiation markers such as BMP2, BMP4, ALP, col1α1, and IGF1. IGFBP5 promotes IGF1 activity, which up-regulates RUNX2 expression. RUNX2 decreases IGFBP2 expression, thus promoting IGF1 action also; (**B**) Schematic magnification of odontogenesis. DPCs (in green) express GHR, IGF-1R, IGF1, and mostly IGFBP4 and IGFBP5 in addition to IGFBP3 and IGFBP2. In the CL, immature proliferating cells express high levels of GHR and IGF-1R. During odontoblast differentiation (yellow cells), GHR and IGF-1R, as well as IGF2 and IGF1, are decreasingly expressed during the whole differentiation process with first IGFBP2 and IGFBP3, then mainly IGFBP5. During amelogenesis, pre-ameloblasts express high levels of GHR and IGF-1R, whose expression decreases during the secretion and the transition stages. During the maturation, corresponding to enamel terminal mineralization, IGF-1R is highly expressed in RA, as well as IGFBP2 and IGFBP3, and to as lesser extend IGFBP5 and IGFBP4. IGFBP6, which inhibits IGF2 action, seems expressed in HERS only (not shown here). ALP: alkaline phosphatase, BMP2, 4: bone morphogenetic protein 2, 4, CL: cervical loop, Col1α1: Collagen 1α1, DPCs: dental pulp cells, GHR: growth hormone (GH) receptor, HERS: Hertwig epithelial root sheath, IGFBP2, 3, 4, 5: insulin-like growth factor-binding protein 2, 3, 4, 5, IGF-1R: insulin-like growth factor 1 (IGF1) receptor RA: ruffle-ended ameloblasts, RUNX2: Runt-related transcription factor 2.

**Figure 3 cells-10-01181-f003:**
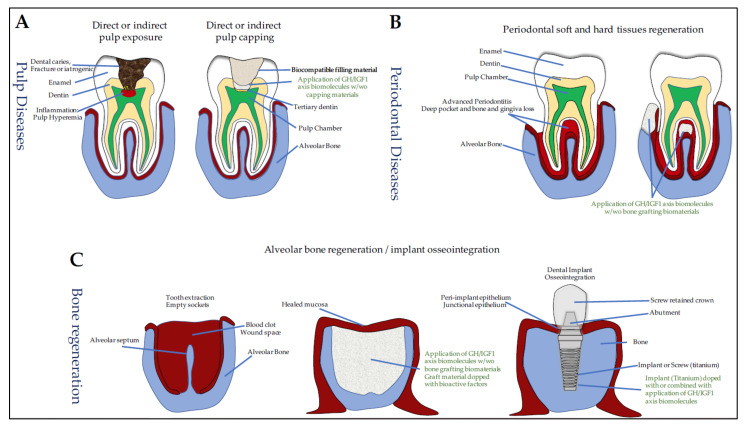
Pathologies of the dento-alveolar complex and therapeutic solutions. (**A**) Dentin and dental pulp diseases with pulp exposures (due to dental decay, fractures, or iatrogenic injuries) activate tertiary dentin synthesis and could be treated with material for pulp capping associated or not with bioactive components of the GH/IGF axis; (**B**) Periodontal diseases (due to infections) are generally associated with gingiva and bone loss need to be filled with biomaterials possibly associated with bioactive components of the GH/IGF axis; (**C**) Implant osseointegration requires alveolar bone regeneration and could be improved with bioactive components of GH/IGF axis. GH: growth hormone, IGF1: insulin-like growth factor 1, w/wo: with/without.

**Figure 4 cells-10-01181-f004:**
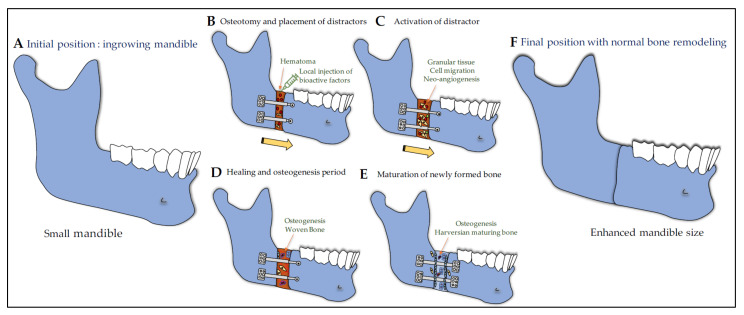
Surgical procedures with rhGH treatments for mandibular size corrections in case of mandibular growth defects. (**A**) Initial position of the ingrowing mandible; (**B**) Osteotomy and placement of distractors; (**C**) Activation of distractors; (**D**) Healing and osteogenesis period; (**E**) Maturation of newly formed bone; (**F**) Final position where normal bone modeling and remodeling are taking place.

## Data Availability

Not applicable.

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
