# Peer review of "The Role of GH/IGF Axis in Dento-Alveolar Complex from Development to Aging and Therapeutics: A Narrative Review"

_cells, 2021, doi:10.3390/cells10051181_

Round 1

Reviewer 1 Report

The topic of the manuscript is the literature review on the role of the GH/IGF axis in the development, ageing and therapy of oral and facial tissues.

The abstract and the main text of the article are informative. The review is interestingly written, including well-prepared figures.

Some following points must be clarified/corrected for the further processing of this manuscript.

Merits-related comments:

  1. The title should be supplemented by the phrase “narrative review”.
  2. Also, keywords should be enriched.
  3. Although the manuscript does not constitute a systematic review, it would be good to add a paragraph on the search and selection of the references included in the review, using part of the PRISMA guidelines (if possible).
  4. Moreover, it is suggested to add more recent articles from 2020-2021 to the references (if available).

Technical comments:

  1. All figures should be inserted into the main text close to their first citation.
  2. Are all figures created by the authors of the manuscript? The annotation in the captions would be useful.
  3. Instead of “6.5.5” should be “6.5.4”.
  4. Instead of “co-workers” better “et al.”.

Reviewer 2 Report

Dear Authors, the submitted manuscript is interesting and prepared with lot of attention. Informations are very valuable. Nevertheless one small paragraph is missing about restorative materials used in dentistry and the response from dental tissues. I also recommend these articles to study: Tooth decay prevalence among children with somatotropin hypopituitarism.   PARTYKA M et al. Bull. Int. Assoc. Paleodontol. 2015 vol. 9 nr 2, 67-72. 

Influence of growth hormone therapy on selected dental and skeletal system parameters.  PARTYKA M et al. Ann. Agric. Environ. Med. 2018 vol. 25 nr 1, 60-65

Reviewer 3 Report

This review is under the scope of this journal; the topic is relevant for readers and this review deals with potentially significant knowledge to the field and an open new way for future studies. The aim of this paper is quite interesting.

However, there are numerous issues in the present manuscript that need to be addressed before publication

  • Some typos in the manuscript (example line 25).

The importance of HERS/ but also the SCAPs

  • Page 3 Line 136 – The Dental papilla, when the root is formed, is in the apical zone, called the papilla apical (stem cells from apical papilla - SCAPs) that remain until the apical closure of the root. Also these, cells can the survival of SCAPs at the infection of the tooth, (in REPs animal study and a Clinical Case, read these references (org/10.3390/APP9193942 and DOI: 10.1016/j.joen.2017.03.005). HERS and the apical papilla are two embryologic structures that coordinate all the radicular development through epithelial-mesenchymal interactions. Apical papilla was a reservoir for mesenchymal stem cells (SCAPs) fundamental for the root development of immature teeth. Normal dentin–pulp complex development requires not only the survival of HERS and ERM but also SCAPs. Endodontic therapies applied to immature teeth that do not affect the apical papilla viability are a key determining factor, guiding successful root development.

  • Expression and action of GH/IGF axis in the BRC and RCRC

Bone remodelling represents the most remarkable bone response to mechanical stress and mineral homeostasis. It is the consequence of complex highly orchestrated and tightly regulated cellular processes taking place in a specialized entity - the Bone Remodelling Compartment (BRC). Please read the Brochado Martins, 2020 (Folia Morphologica Journal), in Remodelling compartment in root cementum (RCRC), Hypothesizing that similar cellular mechanisms underlie bone and cementum remodelling, the present work shows, for the first time, the histological evidence of a specialized remodelling compartment in dental hard tissues.

  • Figures

The figures must be included in the manuscript, and not in section 7. The font in the figure/caption is different from the text. Please, standardized the size and the font in the figures and charts with the font of the manuscript. 

Discussion

  • And also clarified the future perspectives in the discussion. The potential influence of this review on future research plans needs also be discussed.

Round 2

Reviewer 3 Report

This research is under the scope of this journal; the topic is interesting for readers and this research deals with potentially significant knowledge to the field and an open new way for future studies.

The authors improved the quality of the manuscript after the reviewer's indications. Congratulations!